# Targeting *KRAS* G12C Mutation in Colorectal Cancer, A Review: New Arrows in the Quiver

**DOI:** 10.3390/ijms25063304

**Published:** 2024-03-14

**Authors:** Javier Ros, Caterina Vaghi, Iosune Baraibar, Nadia Saoudi González, Marta Rodríguez-Castells, Ariadna García, Adriana Alcaraz, Francesc Salva, Josep Tabernero, Elena Elez

**Affiliations:** 1Medical Oncology, Vall d’Hebron Institute of Oncology (VHIO), 08035 Barcelona, Spaincaterina.vaghi@unimi.it (C.V.);; 2Medical Oncology, Vall d’Hebron Universite Hospital, 08035 Barcelona, Spain; 3Department of Oncology and Hemato-Oncology, University of Milan, 20122 Milan, Italy; 4Department of Hematology, Oncology, and Molecular Medicine, Grande Ospedale Metropolitano Niguarda, 20162 Milan, Italy

**Keywords:** KRAS G12C colorectal cancer, liquid biopsy, mechanism of resistance, KRAS inhibitor, pocket

## Abstract

Kirsten rat sarcoma virus oncogene homolog (*KRAS*) is the most frequently mutated oncogene in human cancer. In colorectal cancer (CRC), *KRAS* mutations are present in more than 50% of cases, and the *KRAS* glycine-to-cysteine mutation at codon 12 (*KRAS* G12C) occurs in up to 4% of patients. This mutation is associated with short responses to standard chemotherapy and worse overall survival compared to non-G12C mutations. In recent years, several *KRAS* G12C inhibitors have demonstrated clinical activity, although all patients eventually progressed. The identification of negative feedback through the EGFR receptor has led to the development of KRAS inhibitors plus an anti-EGFR combination, thus boosting antitumor activity. Currently, several KRAS G12C inhibitors are under development, and results from phase I and phase II clinical trials are promising. Moreover, the phase III CodeBreaK 300 trial demonstrates the superiority of sotorasib-panitumumab over trifluridine/tipiracil, establishing a new standard of care for patients with colorectal cancer harboring *KRAS* G12C mutations. Other combinations such as adagrasib-cetuximab, divarasib-cetuximab, or FOLFIRI-panitumumab-sotorasib have also shown a meaningful response rate and are currently under evaluation. Nonetheless, most of these patients will eventually relapse. In this setting, liquid biopsy emerges as a critical tool to characterize the mechanisms of resistance, consisting mainly of acquired genomic alterations in the MAPK and PI3K pathways and tyrosine kinase receptor alterations, but gene fusions, histological changes, or conformational changes in the kinase have also been described. In this paper, we review the development of KRAS G12C inhibitors in colorectal cancer as well as the main mechanisms of resistance.

## 1. Introduction

Kirsten rat sarcoma virus oncogene homolog (*KRAS*) is the most frequently mutated oncogene in human cancer. In colorectal cancer (CRC), *KRAS* mutations are present in more than 50% of cases [1,2]. The recommended backbone treatment for *KRAS*-mutant colorectal cancer included a fluoropyrimidine-based regimen in combination with oxaliplatin and/or irinotecan and the addition of an anti-VEGF in the first and second line [2]. EGFR inhibitors are not recommended in *RAS*-mutant CRC due to their limited clinical activity. Beyond the second line, trifluridine/tipiracil (-TAS-102-, a combination of trifluridine, a cytotoxic nucleic acid analogue, and tipiracil, a thymidine phosphorylase inhibitor), regorafenib (an oral multikinase inhibitor that blocks VEGFR1-3, KIT, RET, RAF1, BRAF, PDGFR, and FGFR), and recently the combination TAS-102-bevacizumab have demonstrated clinical benefit and are considered the standard of care [3,4,5]. However, not all patients respond to such treatments in the refractory setting and overall survival remains poor. The prognostic impact of a *RAS* mutation is already well-established, demonstrating worse overall survival compared with those tumors with a *RAS/BRAF* wild-type profile [6,7,8,9,10]. In colorectal cancer, the *KRAS* glycine-to-cysteine mutation at codon 12 (*KRAS* G12C) occurs in up to 4% of patients and is associated with poorer overall survival (OS) in the first and second line when treated with chemotherapy. In addition, in the refractory setting among patients treated with TAS-102, *KRAS* G12C mutations were biomarkers for reduced OS benefit from TAS-102 [11,12,13]. Indeed, median overall survival (mOS) among patients with *KRAS* G12C tumors was 16.1 and 9.7 months in the first and second line, whereas mOS for those patients with non-G12C *KRAS*-mutated tumors was 18.3 and 11.4 months in the first and second line, respectively [12,13].

The most frequent *KRAS* mutation in colorectal cancer is the G12D mutation that can be found in up to 42% of colorectal tumors. The G12D mutation has been shown to have an intermediate intrinsic and GAP-mediated GTP hydrolysis rate compared to other G12 and G13 mutants, with mutations such as G12A significantly reducing intrinsic hydrolysis, and G12C exhibiting wild-type levels [14]. *KRAS* G12D mutations have been shown to elicit distinct gene and protein expression profiles compared with other *KRAS* mutations in a tissue-specific manner [15,16].

Importantly, the KRAS G12C protein cycles between an “on” state, in which a guanosine triphosphate (GTP) is attached, and an “off” state in which the GTP loses one phosphate, turning into guanosine diphosphate (GDP). In its active state, KRAS increases downstream oncogenic signaling and cell growth. The G12C mutation impairs GTP hydrolysis, which shifts KRAS to the active GTP-binding state, promoting tumorigenesis and metastases [17,18]. This dynamic conformational change exposes a pocket that can be targeted with specific inhibitors. Indeed, in the last few years, several KRAS G12C inhibitors, including adagrasib, sotorasib, and divarasib among others, have been developed as monotherapy or in combination with anti-EGFR agents, demonstrating meaningful clinical activity. New combinations that boost antitumor efficacy and overcome acquired resistance are currently being developed. In this paper, we review the development of KRAS inhibitors in metastatic colorectal cancer, the already described mechanism of resistance, and novel combinations to overcome such resistances and boost the antitumoral effect.

## 2. The *RAS* Pathway and Downstream Signaling

RAS proteins are small, membrane-bound guanine nucleotide-binding proteins involved in several signaling pathways that ultimately regulate cell growth, motility, angiogenesis, and survival in various cancer types. They are associated with tumor progression and resistance to targeted therapies. In physiological conditions, RAS shifts between two conformational states: an active state, where an active guanosine-5′-triphosphate (GTP) is attached to the RAS protein, and an inactive state, where GTP is processed to become guanosine diphosphate (GDP). The initial step in RAS activation involves the activation of several receptor tyrosine kinases (RTKs), induced by ligand binding in the extracellular domain of RTKs. This leads to RTK dimerization and autophosphorylation. The activated receptors then interact with the growth factor receptor protein 2 (GRB2), recruiting guanine nucleotide exchange factors (GEFs), including the Son of Sevenless homolog (SOS). SOS promotes GDP/GTP exchange, inducing a conformational change that activates the kinase. This activation triggers different pathways, including RAF/MEK/ERK, PI3K/AKT/mTOR, and nuclear transcription factors involved in cell survival and metastases. The RAS cycle is ultimately switched off by GTPase-activating proteins (GAPs), which induce GTP hydrolysis, forming inactive RAS-GDP. The G12C mutation occurs at the 12th position of the *KRAS* gene, where glycine (G) is replaced by a cysteine (C) amino acid [17,18]. This mutation leads to the production of a mutated KRAS protein with altered function, which remains constitutively active. In this case, an allosteric pocket below the switch II region of the mutant cysteine was identified by Ostrem et al. [19]. G12C inhibitors preferentially bind RAS in the GDP-bound conformation, blocking the exchange with GTP and thus preventing the activation of the signaling cascade. Figure 1 pictures Ras signaling pathways and drugs associated with KRAS G12C inhibition.

## 3. Development of KRAS G12C Inhibitors

The emergence of KRAS G12C inhibitors represents a new therapeutic strategy in this population. In addition to the development of several inhibitors, upfront and acquired mechanisms of resistance have been identified and some strategies have been developed to overcome such resistance.

### 3.1. KRAS Inhibitors in Monotherapy

#### 3.1.1. Sotorasib

After the discovery of compounds that covalently bind to the switch II pocket of KRAS G12C in its inactive GDP-bound state (off) and the understanding of the mechanism of KRAS G12C inhibition, sotorasib (AMG510) was developed and entered the clinical setting in 2018 as the first-in-class anti-KRAS G12C [19,20,21,22,23]. In the phase I/II CodeBreaK 100 trial, 129 patients with previously treated advanced solid tumors harboring the *KRAS* G12C mutation were enrolled in the phase I cohort [24]. Among them, 42 were heavily pretreated. In total, twenty-five patients received the expansion dose of 960 mg qd (from the Latin “quaque die”, once a day), which was subsequently established as the recommended phase II trial dose (RP2D). No dose-limiting toxic effects occurred, nor did any treatment-related adverse event resulting in death. The most common treatment-related adverse events (TRAEs) were diarrhea, fatigue, nausea, and an increase in aspartate or alanine aminotransferase, occurring in 30%, 23%, 21%, 13%, and 12% of the total population, respectively. Regarding anti-tumor activity, three out of 42 mCRC patients (7.1%) responded, with all responses occurring in the cohort treated with the 960 mg qd dose. The disease control rate (DCR) was 73.8% in the total mCRC patients and 80% in the 960 mg qd cohort. The subsequent phase II cohort of the CodeBreaK 100 trial included 62 patients with a mean of three previous lines [25]. These patients received sotorasib 960 mg qd. After almost 2 years, with five patients still on the treatment, the objective response rate (ORR) was 9.7%, with tumor shrinkage in 66% of patients and a DCR of 82% of patients. The median progression free survival (mPFS) was 4 months. Grade 3 TRAEs occurred in six patients (10%); diarrhea, fatigue, nausea, and elevation of transaminases were confirmed as the most common. The only grade 4 event observed was an increase in blood creatinine phosphokinase. Apart from being safe and tolerable, sotorasib monotherapy demonstrated only modest anti-tumor activity in mCRC, especially if compared with other histologies such as NSCLC, in which sotorasib monotherapy achieved an ORR of 41% and a mPFS of 6.3 months [26]. Table 1 summarizes clinical trials evaluating KRAS G12C inhibition in patients with *KRAS* G12C-mutated colorectal cancer.

#### 3.1.2. Adagrasib

The KRAS G12C inhibitor adagrasib (MRTX849) binds irreversibly and selectively to KRAS G12C in its inactive state [27]. In the phase I/Ib KRYSTAL-1 trial, where 25 patients with advanced, previously treated, solid tumors harboring *KRAS* G12C mutations were enrolled and treated with various expansion doses, four of them were affected with mCRC, with two receiving the RP2D of 600 mg bid (from the Latin “bis in die”, two times a day). A toxicity profile consistent with sotorasib was observed, with nausea (80.0%), diarrhea (70.0%), vomiting (50.0%), and fatigue (45.0%) being the most prevalent TRAEs in the cohort treated with the RP2D. The most common grade 3/4 TRAE was fatigue (15.0%). In terms of pharmacokinetics, adagrasib displayed a longer half-life compared to sotorasib (23 h versus 5.5 h), along with a higher exposure dose dependency and ability to penetrate the central nervous system. Notably, one out of the two mCRC patients treated with the RP2D exhibited a partial response (PR), with a DOR lasting 4.2 months. Additional efficacy data came from the phase I/II trial KRYSTAL-1, in which 44 patients affected with mCRC were treated with adagrasib 600 mg bid, with 19% achieving PR, an mDOR of 4.3 months, and an mPFS of 5.6 months [28].

#### 3.1.3. Divarasib

The clinical development of divarasib (GDC-6036) started in 2020. Compared to its predecessors, divarasib boasts a potency of five to twenty times greater and a selectivity of up to 50 times higher [29]. Notably, at equivalent nanomolar concentrations, divarasib achieves a higher relative alkylation of KRAS G12C compared to sotorasib and adagrasib. Furthermore, non-KRAS G12C proteins are alkylated at higher concentrations with divarasib compared to the other two drugs. In a phase I trial assessing the safety of divarasib, out of the 137 patients enrolled, 55 were diagnosed with mCRC [30]. Grade 3 or higher TRAEs were observed in 12% of the total population, including diarrhea, increased aspartate aminotransferase, nausea, vomiting, and fatigue. A long half-life of 17 h was observed. Regarding efficacy in the mCRC population, a complete response (CR) or PR was observed in 29% of patients, including one CR. Importantly, higher ORR and mPFS of 36% and 6.9 months, respectively, emerged when analyzing only the subset of 39 patients treated with the RP2D of 400 mg qd. All partial responses were accompanied by a reduction in circulating tumor DNA (ctDNA) *KRAS* G12C allele frequency to less than 1% after 2 cycles. Co-occurring mutations in *APC* and *TP53* did not significantly impact the response. Analysis of pre- and post-treatment ctDNA from 16 responders revealed that at least one genomic alteration mechanistically linked to KRAS inhibitor resistance developed in nine patients. These alterations included *RAS* copy number gain or amplification, non-*KRAS* G12C mutations, alterations in receptor tyrosine kinases (RTKs) and mitogen-activated protein kinase (MAPK) pathway components, including *ERBB2* and *MET* amplifications, and *BRAF* mutations, respectively. Conversely, at least one pre-existing mutation in all *RAS* genes was found in six out of 25 patients in the total population with progressive disease (PD) as the best response, including one mCRC patient, with an adaptive increase and decrease in the allele variant frequency of the non-*KRAS* G12C and *KRAS* G12C mutations, respectively, in three out of these six. Again, a striking difference in efficacy endpoints between NSCLC and CRC was observed with divarasib, as higher ORR of 53.4% and mPFS of 13.1 months were achieved in NSCLC patients.

#### 3.1.4. Other Inhibitors

The aggregate data of two phase I trials (NCT05005234, NCT05497336) in which Fulzerasib (IBI351) has been tested in 45 mCRC patients with *KRAS* G12C mutations have been reported. Among the 32 evaluable patients, an ORR of 43.8% (14/32) was observed. Grade 3 TRAEs occurred in nine patients (20.0%), with no drug-related adverse events leading to treatment discontinuation or death [31]. Finally, Garsorasib (D-1553) monotherapy demonstrated deep clinical activity and safety in heavily pretreated patients with *KRAS* G12C-mutated CRC. Based on this initial activity, the combination of Garsorasib and cetuximab was evaluated in a phase II trial [32]. This trial included 40 patients, 80% of them having received at least two previous lines. ORR was 45% with a DCR of 95%. mPFS was 7.6 months whereas OS data is still immature. No grade 4/5 TRAEs were reported [29].

### 3.2. A Step Forward in Boosting Antitumor Activity: Combining KRAS G12C Inhibitors with Anti-EGFR

From the outset, a notable discrepancy in efficacy emerged between patients with NSCLC and CRC in the early phases of trials involving sotorasib. Impressively, the best ORR reported was 41% in NSCLC compared to 9.7% in mCRC (CodeBreaK 100) [25,26]. Amodio et al. uncovered several variances at the cellular level, notably observing that colorectal cancer lines exhibited a higher basal level of phosphorylated-functional-receptor tyrosine kinases (RTKs) and maintained responsiveness to further activation of EGFR by growth factors [33]. In this milieu of concurrent mechanisms, inhibiting *KRAS* G12C can not result in quelling cell growth and proliferation. Moreover, a more pronounced rebound of the MAPK pathway following *KRAS* G12C inhibition was demonstrated in CRC compared to NSCLC cell lines. Consequently, the synergistic combination of *KRAS* G12C inhibitors and anti-EGFR therapies was investigated both in vitro and in colorectal patient-derived models, yielding significant tumor regression or even complete remission. Rayan et al. further explored adaptive resistance to *KRAS* G12C inhibitors, identifying the rebound of the MAPK pathway as a principal mechanism [34]. This rebound, characterized by marked induction of GTP-bound forms of wild-type HRAS and KRAS, which was otherwise abrogated by the knockdown of HRAS and NRAS, leads to MAPK pathway activation in a *KRAS* G12C-independent way, suggesting that it can not be overcome by escalating the dosage of *KRAS* G12C inhibitors. Instead, a strategy involving upstream and downstream inhibition, specifically with SHP2 or MEK inhibitors, resulted in a robust and sustained inhibition of MAPK signaling in vitro. This study suggested an alternative approach, consisting of a combination of *KRAS* G12C inhibitors with compounds targeting RTK signaling. These findings paved the way for the combination of KRAS G12C inhibitors with either panitumumab and cetuximab, and with other inhibitors along the MAPK pathway, which are currently under investigation in combination with *KRAS* G12C inhibitors in ongoing clinical trials [18,35].

#### 3.2.1. Adagrasib Cetuximab

In the phase I/II trial KRYSTAL-1, adagrasib was also tested in combination with cetuximab in 32 patients [28]. Grade 3 or higher TRAEs were reported in 34% of the monotherapy group and 16% of the combination therapy group, with no grade 5 adverse events observed. However, 16% of patients receiving cetuximab discontinued the drug due to toxicity. The toxicity profile of the combination only differed in cetuximab-related adverse events, without any synergistic toxic effects observed. Notably, in patients receiving adagrasib with cetuximab, the ORR increased from 19% to 46%, along with a mPFS of 6.9 months, compared to adagrasib monotherapy. An exploratory analysis of ctDNA revealed a higher clearance of the *KRAS* G12C mutant allele after two cycles in the combination therapy group compared to the monotherapy group (88% vs. 55%), correlating with the different ORR. No association between response and *PI3KCA* and *TP53* mutations were identified. The phase III KRYSTAL-10 trial comparing adagrasib plus cetuximab vs. second-line chemotherapy has completed recruitment.

#### 3.2.2. Sotorasib Panitumumab

The combination of sotorasib and panitumumab has also been investigated [36,37]. The phase Ib trial, CodeBreaK 101, included cohorts specifically focused on mCRC patients, comprising a dose exploration cohort of eight patients to determine the RP2D of sotorasib in combination with panitumumab and a dose expansion cohort that included 40 patients [36]. In total, 27% of patients experienced TRAEs of grade ≥3, mostly attributed to panitumumab, including rash (6%), acneiform dermatitis (4%), and hypomagnesemia (4%). Panitumumab led to interruption or reduction more often than sotorasib (29% versus 15%), but no discontinuation of any drug due to a treatment-related adverse event was reported. In the exploratory cohort, which included 63% of patients previously treated with sotorasib monotherapy, ORR was 12% and the DCR was 80%. Conversely, in the expansion cohort (anti-KRAS naive), ORR and DCR were higher at 30% and 92%, respectively, with a mPFS of 5.7 months. Importantly, the study also evaluated the impact of sidedness but no difference in terms of response based on primary tumor location was observed.

The phase III trial CodeBreaK 300 tested sotorasib in the refractory setting at two different doses (960 mg and 240 mg qd), both in combination with panitumumab, and compared the results with the standard of care, consisting of either regorafenib or trifluridine/tiparacil [37]. Despite the RP2D having already been set at 960 mg qd, the alternative dose of 240 mg qd of sotorasib was tested due to its non-linear pharmacokinetic properties. The trial reached its primary endpoint by demonstrating the superiority of the combination of panitumumab and sotorasib at the two doses tested compared to the SOC in terms of PFS [5.7 versus 2.2 months, HR 0.49 (95% CI, 0.30 to 0.80; *p* = 0.006) and 3.9 versus 2.2, HR 0.58 (95% CI, 0.36 to 0.93; *p* = 0.03), respectively]. A significant difference in terms of ORR also emerged among the arms, in which CR/PR was achieved in 26%, 5.7%, and 0% of patients treated with sotorasib 960 mg qd and panitumumab, sotorasib 240 mg qd and panitumumab, and regorafenib or trifluridine/tiparacil, respectively. Data on overall survival is still immature. Regarding treatment-related events, a grade 3 or higher TRAE occurred in 35.8%, 30.2%, and 43% of patients treated with the combination of sotorasib and panitumumab at a dose of either 960 mg qd or 240 mg qd, respectively, and with the SOC. The most common grade 3 or higher TRAEs of regorafenib-trifluridine/tiparacil were neutropenia (23.5%), anemia (5.9%), and hypertension (5.9%), whereas for the combinations, the grade 3 or higher TRAEs were panitumumab-related dermatitis acneiform (11.3%), hypomagnesemia (5.7%), and rash (5.7%). Finally, sotorasib and panitumumab were also investigated in combination with FOLFIRI in 33 pretreated mCRC with at least one prior line of systemic therapy. ORR was 58%. More grade 3 or higher TRAEs were observed, up to 45%, mostly dermatologic [38].

#### 3.2.3. Divarasib Cetuximab

Divarasib was also tested in combination with cetuximab in a cohort of 29 patients with *KRAS* G12C mutation mCRC, predominantly administrated at the dose of 400 mg qd [39]. Patients had been previously treated with at least two prior lines of systemic therapy, including KRAS G12C inhibitors in five patients. A similar frequency of grade 3 or higher TRAEs was observed with the combination and monotherapy (11%). However, a more complex toxicity profile emerged, as in addition to diarrhea, lipase elevation, hypomagnesemia, rash, and other anti-EGFR-related toxicities also occurred. Two grade 4 TRAEs were reported: one case of hypomagnesemia and one neutropenia. Among patients who have not previously received KRAS G12C inhibitors, the ORR was 62.5% and the mPFS was 8.1 months. Interestingly, three out of five patients pre-treated with KRAS G12C inhibitors achieved PR. CtDNA analysis revealed a significant and widespread decline in *KRAS* G12C variant allele frequency (VAF), with a KRAS G12C VAF below 0.5% after two cycles in 77% of patients, including one patient achieving stable disease (SD). At the time of disease progression, an acquired genomic alteration potentially related to adaptive resistance was identified in 13 out of 14 profiled patients; genomic alterations were consistent with those reported in patients receiving divarasib monotherapy.

### 3.3. Ongoing Clinical Trials and New KRAS G12C Inhibitors in Colorectal Cancer

Several clinical trials investigating the safety and efficacy of both well-established and new KRAS G12C inhibitors, alone, in combination with anti-EGFR, or combined with inhibitors of other upstream/downstream modulators of the MAPK pathway, are ongoing.

A combination of adagrasib and cetuximab is under further investigation in a phase III trial that compares the combination to chemotherapy in a second-line setting (KRYSTAL-10). The addition of irinotecan to adagrasib and cetuximab is under evaluation in a phase I trial, to explore the safety and activity of the combination (NCT05722327). Boosting KRAS G12C inhibition through targeting upstream/downstream effectors of the MAPK pathway is also under evaluation. The main MAPK pathway effectors targeted by combination compounds under evaluation include SHP2 (Src homology region 2-containing protein tyrosine phosphatase 2), a non-receptor-type protein tyrosine phosphatase encoded by the gene *PTPN11* that acts downstream of most receptor tyrosine kinases, and SOS1 (Son of Sevenless homolog 1), a GEF that switches RAS-GDP to its active conformational status. Simultaneous inhibition of SOS1 and KRAS, which was synergistic in preclinical models of *KRAS* G12C-mutated cancer cells [40], is now under clinical investigation in KRYSTAL-14 and other trials (such as NCT05578092). Similarly, the KRYSTAL-2, KontRASt-1, NCT05288205, and NCT06024174 are currently evaluating concurrent inhibition of KRAS and SHP2, as it boosted KRAS inhibition in preclinical models [34,41,42,43]. In addition, adagrasib is also under investigation in combination with the CDK4/6 (cycline-dependent kinase 4/6) inhibitor palbociclib in KRYSTAL-16, as it boosts KRAS inhibition due to the fact that KRAS is known to mediate cell proliferation partly through the Cyclin D family [44]. The concurrent inhibition of KRAS and CDK4/6 demonstrated increased antitumor activity in preclinical models [45]. Another putative target of combinational inhibition under clinical investigation in combination with garsorasib is the downstream effector of the MAPK pathway FAK (focal adhesive kinase), a non-receptor tyrosine kinase that promotes tumorigenesis. In fact, its concurrent blockade was found superior to anti-KRAS monotherapy in preclinical models [46].

The combination of divarasib and cetuximab, plus or minus chemotherapy, is under investigation in the INTRINSIC trial (NCT04929223), an umbrella trial evaluating the safety and efficacy of diverse targeted therapies in specific subpopulations of patients with mCRC, including those harboring *KRAS* G12C.

Several other anti-KRAS G12C compounds have been developed and are now being tested in phase I trials, mainly in combination with anti-EGFR therapy, anti-SHP2, anti-SOS1, or a pan RAS inhibitor. Pharmacodynamic and pharmacokinetic differences between these compounds might translate into differential efficacy. For example, JDQ443 and JNJ-74699157 interact with a different cysteine residue than other KRAS G12C inhibitors [47]. Thus, this suggests that they may be able to overcome mechanisms of resistance consisting of target mutations that have already been reported [47,48,49]. Conversely, RMC 6291 has a unique and innovative mechanism of action compared to other KRAS G12C inhibitors: it binds to cylophilin A and forms a binary complex which then blocks KRAS G12C in its GTP-bound state (on) in a tertiary complex, leading to the disruption of RAS effector binding and direct extinction of KRAS G12C (on) signaling [50]. Although it outperformed other anti-KRAS G12C (off) inhibitors in ex vivo models, clinical data are still awaited. Table 2 summarizes ongoing clinical trials evaluating KRAS G12C inhibition in patients with *KRAS* G12C-mutated colorectal cancer, and Table 3 summarizes the most relevant differences between adagrasib, sotorsaib, and divarsib.

## 4. Mechanisms of Resistance

### 4.1. EGFR-Mediated Adaptive Feedback Reactivation of the RAS-MAPK Pathway

The first clinical results with KRAS G12C inhibitors in monotherapy suggested that the KRAS G12C inhibition was lineage-specific. Colorectal cancer cell lines have high basal RTK activation compared to NSCLC cell lines, and in colorectal cancer, G12C inhibition promotes higher phospho-ERK rebound than in NSCLC. Amodio et al. identified EGFR signaling as the main mechanism of resistance to KRAS G12C inhibitors, paving the way to a new strategy of combinatorial targeting of both EGFR and KRAS G12C. This was proven to be highly effective in patient-derived organoids and xenografts [33]. Figure 2 summarizes the mechanisms of resistance and proposed therapeutic strategies to overcome it.

### 4.2. Acquired Genomic Events

From the results of paired plasma sample analyses from patients treated with divarasib +/− cetuximab, up to 90% had at least one acquired genomic alteration associated with treatment resistance [30,39]. Most common genomic mechanisms of resistance included genomic *KRAS* alterations (including mutations in non-G12C *KRAS*, *BRAF*-V600E, *HRAS*, *MAP2K1*, and *KRAS*, *NRAS*, and *BRAF* amplifications) that led to *KRAS* oncogenic activation, but also alterations in the PI3K and RTK pathways components, including *EGFR*, *MYC*, and *MET* amplifications and *ALK* and *RET* fusions [30,39]. Similar data has been reported with adagrasib +/− cetuximab, where genomic acquired mechanisms of resistance were detectable in more than 70% of the patients, mostly affecting genes coding for MAPK and PI3K pathway effectors and RTKs [51]. Importantly, responses were observed regardless of EGFR expression. Among patients treated with sotorasib monotherapy, genomic acquired mechanisms were also common (71%), and mostly involved RTK genes (27%), above all *EGFR*, *ERBB2*, and *KIT* [52]. In fact, more than 30% of the patients had more than three genomic mechanisms of resistance. New RTK alterations frequently emerged at progression in CRC, highlighting the potential role of combining KRAS G12C inhibitors with upstream inhibitors such as SHP2 or EGFR inhibitors. Finally, deep mutational scanning (in silico) demonstrates drug-specific patterns of resistance, with some mutations conferring resistance to adagrasib but not to sotorasib, and vice versa [49]. These pieces of evidence demonstrate the distinctiveness and complexity of the secondary resistance to KRAS G12C inhibition and pave the way to overcoming it through the development of both novel KRAS inhibitors with alternative modes of binding and different allele specificities, and effective combination therapy regimens with inhibitors of other effectors in the MAPK pathway. Both these strategies are already under investigation [41,47,48,50]. On the contrary, besides promising pre-clinical evidence of effective concurrent blockading of PI3K/AKT/mTOR signaling, no clinical trial is currently ongoing to our knowledge [43,53].

### 4.3. KRAS Switch-II Pocket Mutations

Most of the patients treated with KRAS G12C inhibitors will develop polyclonal genomic alterations as a central acquired mechanism of resistance. Notably, several *KRAS* mutations including R68S, H95R, and Y96D mutations affect the switch-II pocket, to which KRAS G12C inhibitors bind, conferring resistance to these drugs [49,54]. In patient-derived *KRAS* G12C models, a novel functionally distinct tricomplex KRAS G12C active-state inhibitor (RM-018) has been tested preclinically and successfully inhibited KRAS G12C/Y96D, thus overcoming this mechanism of resistance. Collectively, mutations that disrupt covalent drug binding can lead to clinical resistance to KRAS G12C inhibitors.

### 4.4. Histological Switch

In two out of ten patients (one patient with CRC and the other with non-small cell lung carcinoma) treated with adagrasib monotherapy, from whom paired tumor tissue was available, the histologic transformation from adenocarcinoma to squamous cell carcinoma was observed without any identifiable genomic mechanism of resistance [49]. Studies in G12C and G12D *KRAS*-mutant lung cancer mouse models and organoids treated with KRAS inhibitors reveal that tumors invoke a lineage plasticity program to switch from adenocarcinoma to squamous via transcriptomic and epigenomic changes, modulating the response to KRAS inhibition [55].

## 5. Discussion

*KRAS* G12C mutations are present in a small percentage of metastatic colorectal cancer patients [9]. However, it is associated with a poor response to standard treatments and shorter overall survival compared with non-G12C mutations [12,13]. Thus, the development of targeted agents that can improve these outcomes is paramount. In the last decade, several KRAS G12C inhibitors have emerged. Nonetheless, the understanding of the underlying biology of these tumors has also been crucial in identifying and overcoming mechanisms of resistance. Early on, after the first patients were treated with KRAS G12C inhibitors, it was observed that, like *BRAF*-mutated tumors [56,57], when blocking the *KRAS* mutation, negative adaptive feedback through the EGFR receptor emerges, leading to resistance [33]. Thus, subsequent studies included either cetuximab or panitumumab to overcome this mechanism of resistance. Indeed, with this combination, clinical outcomes improved as the drugs boost the antitumor effect. In this scenario, adagrasib and sotorasib are the first inhibitors being developed and the ones that are in a more advanced stage. The CodeBreaK 300 trial, combining sotorasib and panitumumab, has been the first phase III randomized trial to demonstrate the benefit of a KRAS inhibitor over standard treatment in colorectal cancer [38]. Currently, the KRYSTAL-10 trial, a randomized phase III trial comparing adagrasib plus cetuximab vs. chemotherapy, has finished recruitment, and results are awaited in the upcoming months. In addition, new KRAS inhibitors such as divarasib or garsorasib have also demonstrated deep clinical activity not only in monotherapy, with disease control rates of 84% and 95%, respectively, but also in combination with anti-EGFR agents. When divarasib was combined with cetuximab, the DCR increased to 95.8% [30,39]. However, data from these trials, despite being promising, need to be carefully interpreted as they come from small, non-randomized phase I and phase II clinical trials. However, it could be expected that the combination of these new KRAS inhibitors plus anti-EGFR may improve the patients’ clinical outcomes significantly. Regarding treatment-related adverse events, the toxicity profile is similar and manageable among different KRAS G12C inhibitors, mostly low-grade gastrointestinal adverse events and rash.

On the other hand, despite the addition of anti-EGFR to overcome the EGFR-mediated adaptive feedback reactivation of the RAS-MAPK pathway following KRAS G12C inhibition, eventually all patients will progress. In this regard, liquid biopsy has been demonstrated to be a reliable tool in colorectal cancer to monitor response, to forecast prognosis, and to identify genomic mechanisms of resistance [58,59,60,61,62,63]. Circulating tumor DNA from the KRYSTAL-1 and CodeBreaK 100 trials has been demonstrated to be able to identify multiple acquired pathological genomic alterations among patients treated with KRAS G12C inhibitors, with or without anti-EGFR, in up to 70% of the patients, but particularly among patients treated with the combination [51,52]. In addition, through ctDNA, it has been identified that patients receiving the double combination have larger decreases in ctDNA, and this was associated with deeper responses. Indeed, those patients with a mutant allele fraction clearance (MAFC) higher than 90% achieve an ORR of 47% and 67% with adagrasib and adagrasib-cetuximab, respectively, whereas patients with a MAFC lower than 90% have an ORR of 8% and 33% with adagrasib and adagrasib-cetuximab, respectively [51]. Similar results were also observed in phase 1b, where divarasib and cetuximab were combined, in which patients with higher MAFC achieve higher ORR compared with patients with lower MAFC [39]. Based on these results, it is clear that ctDNA can be used to track tumor dynamics and identify mechanisms of resistance. In this regard, it could be hypothesized that, similar to the *RAS* wild-type and *BRAF* story in which rechallenge (anti-EGFR and BRAF inhibitor plus anti-EGFR, respectively) guided with liquid biopsy has been demonstrated to improve clinical outcomes [64,65,66,67]. If there is a resistance to clonal decay, perhaps the reintroduction of KRAS G12C inhibitors may be effective in a well-selected population. However, liquid biopsy has also highlighted that RTK alterations are commonly found in this scenario, which opens a window of opportunity to test new drugs such as SHP2 inhibitors or combinations with immune checkpoint inhibitors. Lastly, less common but also important mechanisms of resistance need to be discussed, such as the histological switch after a KRAS inhibitor-based combination or the conformational changes of the switch-II pocket produced by specific acquired *KRAS* mutations. The first scenario highlights the importance of obtaining, when possible, tumor tissue upon progression, as this tissue can be informative in helping make decisions such as changes to a specific chemotherapy regimen based on histological differentiation. The second scenario is also a window of opportunity to test new tricomplex inhibitors that can block KRAS activation even when conventional inhibitors are not able to because of three-dimensional structural changes. Finally, the combination of chemotherapy with KRAS G12C inhibitors has also demonstrated deep clinical responses. The phase 1b CodeBreaK 101 subprotocol H is currently evaluating the combination of FOLFIRI plus sotorasib plus panitumumab in the refractory setting with an ORR of 58% and a DCR of 93%, including patients previously treated with KRAS G12C inhibitors, thus suggesting that rechallenge with KRAS G12C inhibitors in combination with chemotherapy may lead to responses even among patients who have previously progressed to a KRAS G12C inhibitor [38].

Finally, many studies have pointed out the role of the gut microbiome as a prognostic and predictive biomarker in colorectal cancer [68,69]. In colorectal cancer, *Fusobacterium nucleatum* has been related to genetic and epigenetic lesions, such as microsatellite instability, the CpG island methylator phenotype, and genome mutations in colorectal cancer tissues [70]. Indeed, *F. nucleatum* could promote proliferation and metabolism, remodel the immune microenvironment, and facilitate metastasis and chemoresistance in the tumorigenesis and development of CRC [68]. In addition, *F. nucleatum* is enriched in *KRAS* G12-mutant CRC tumor tissues and contributes to colorectal tumorigenesis. Thus, personalized modulation of the gut microbiota may provide a more targeted strategy for CRC treatment.

All in all, the landscape of KRAS G12C inhibitors in colorectal cancer is changing faster, and the identification of mechanisms of resistance has been paramount in boosting the antitumor activity of these combinations. Despite data from randomized phase III trials still being scarce, data from phase II trials evaluating new molecules are promising.

## 6. Conclusions

Even though *KRAS* G12C mutations represent a small percentage of all mCRC (around 4%), the development of KRAS G12C inhibitors has demonstrated meaningful clinical activity. Currently, there is only one randomized phase III trial demonstrating benefit over trifluridine/tipiracil, but ongoing trials, combining KRAS inhibitors with anti-EGFR, show promising results and may increase therapeutic options in different settings in the upcoming years. In addition, the identification of mechanisms of resistance has led to new treatment strategies to overcome acquired mechanisms of resistance such as conformational changes of the switch-II pocket, acquired genomic events, or histological changes. In this scenario, liquid biopsy may help to track tumor responses, identify mechanisms of resistance, and track clonal decay to develop new treatment strategies. There are new arrows in the quiver for *KRAS*-mutated colorectal cancer.

## Figures and Tables

**Figure 1 ijms-25-03304-f001:**
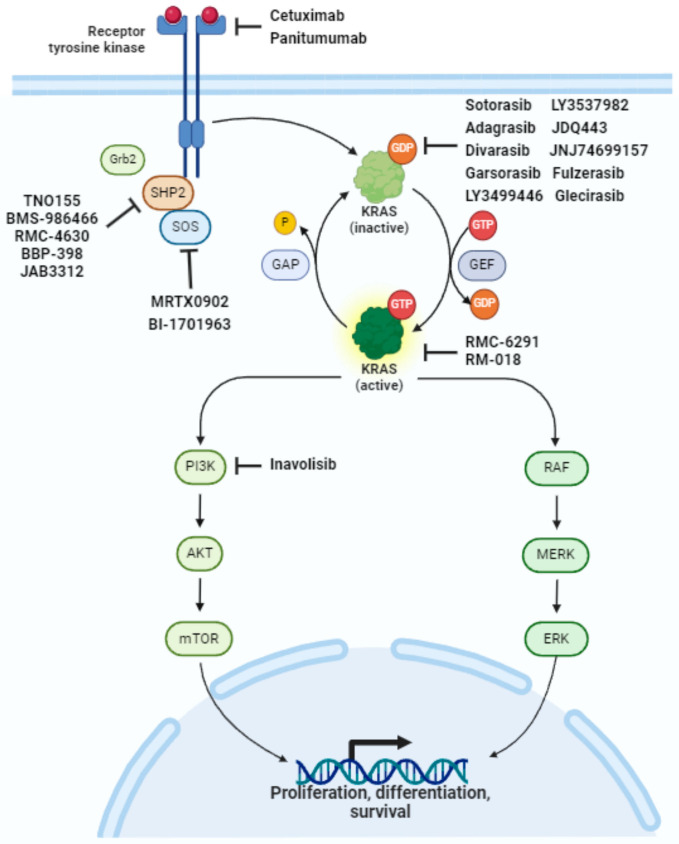
RAS signaling pathways and drugs associated with KRAS G12C inhibition.

**Figure 2 ijms-25-03304-f002:**
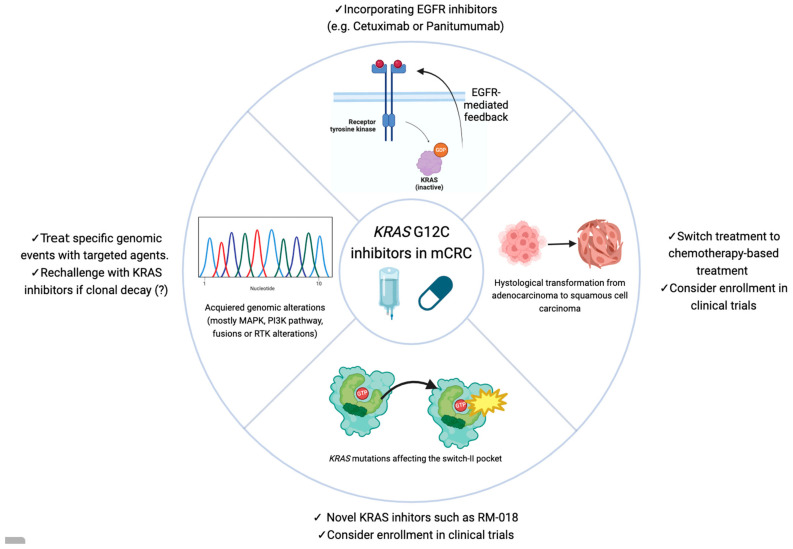
Summary of mechanisms of resistance and proposed therapeutic strategies to overcome it.

**Table 1 ijms-25-03304-t001:** Key clinical trials targeting KRAS G12C in mCRC, completed or with already published data.

Clinical Trials Targeting KRAS G12C in mCRC, Completed or with Already Published Data				
Study Name/ID	Population (n. of Patients)	Treatment Regimen (n. of Patients Treated)	Results	Grade 3 or Higher TRAEs
Phase				
CodeBreaK 100/NCT03600883	advanced *KRAS* G12C mutant solid tumors (124, including 42 mCRC)	Sotorasib (AMG510)	limited to mCRC treated with any dose	52.7%, in the overall population
Phase I			ORR 7.1% (3/42)	
			DCR 73.8% (31/42)	
			mPFS 4 mo	
CodeBreaK 100 (CRC expansion cohort)/NCT03600883	advanced *KRAS* G12C mutant mCRC (62)	Sotorasib (AMG510) 960 mg qd	ORR 9.7% (6/62)	10% (6/62)
			DCR 82.3% (51/62)	
Phase II			mPFS 4 mo	
KRYSTAL-1/NCT03785249	advanced *KRAS* G12C mutant solid tumors (25, including 4 mCRC)	Adagrasib (MRTX849)	limited to evaluable mCRC treated with 600 mg bid	36% (9/25)
Phase I/Ib			ORR 50% (1/2)	
			DOR 4.2 mo	
KRYSTAL-1 (monotherapy arm)/NCT03785249	advanced *KRAS* G12C mutant mCRC (44)	Adagrasib (MRTX849) 600 mg bid	ORR 19% (8/43)	34% (15/43)
Phase I/II			DCR 86% (37/43)	
			mPFS 5.6 mo	
NCT04449874	advanced *KRAS* G12C mutant solid tumors (137, including 55 mCRC)	Divarasib (GDC-6036)	limited to mCRC population	7% (4/55)
Phase Ib			ORR 29.1% (20/55)	
			mPFS 5.6 mo	
			limited to mCRC treated with 400 mg qd	
			ORR 35.9% (14/39)	
			mPFS 6.9 mo	
pooled analysis of NCT05005234 and NCT05497336	advanced *KRAS* G12C mutant solid tumors, including 45 mCRC	Fulzerasib (IBI531)	limited to mCRC patients treated with 600 mg bid	20% (9/32)
Phase I			ORR 43.8% (14/32)	
			DCR 87.5% (28/32)	
Clinical trials targeting EGFR-KRAS G12C in mCRC, completed or with already published data				
CodeBreaK 101/NCT04185883	advanced *KRAS* G12C mutant mCRC (48)	Sotorasib (AMG510) + panitumumab	limited to patients treated with 960 mg qd	27% (13/48)
Phase Ib			ORR 30% (12/40)	
			DCR 92.5% (37/40)	
			mPFS 5.7 mo	
CodeBreaK 101 (subprotocol H)/NCT04185883	advanced *KRAS* G12C mCRC previously treated ≥1 prior treatment (33)	Sotorasib (AMG510) 960 mg qd + panitumumab + FOLFIRI (54)	ORR 58.1%	45.5% (15/33)
			DCR 93.5%	
Phase Ib			mPFS 5.7 mo	


CodeBreaK 300/NCT05198934	advanced *KRAS* G12C mutant mCRC (160)	Sotorasib (AMG510) 960 qd mg + panitumumab (53)	ORR 26.4% DCR 71.7%	35.8% (19/53)
Phase III			mPFS 5.6 mo	
		Sotorasib (AMG510) 240 qd mg + panitumumab (53)	ORR 5.7% DCR 67.9%	30.2% (16/53)
			mPFS 3.9	
		SOC (54)	ORR 0% DCR 46.3%	43.1% (23/54)
			mPFS 2.2 mo	
KRYSTAL-1 (combination arm)/NCT03785249	advanced *KRAS* G12C mutant mCRC (32)	Adagrasib (MRTX849) 600 mg bid + cetuximab	ORR 46% (13/28)	16% (5/32)
Phase I/II			DCR 100% (28/28)	
			mPFS 6.9 mo	
			mOS 13.4 mo	
NCT04449874 (arm C)	advanced *KRAS* G12C mutant mCRC (29)	Divarasib (GDC-6036) at 400 mg qd (26) + cetuximab	limited to KRAS G12C inhibitor naive population	37.9% (11/29)
Phase Ib			ORR 62.5% (14/24)	
			mPFS 8 mo	
NCT04585035	advanced *KRAS* G12C mutant solid tumors, including 29 mCRC	Garsorasib (D-1553) 600 mg bid (29) + cetuximab	ORR 51.7% (15/29)	10.3% (3/29)
Phase I/II			DCR 93.1% (27/29)	
			mPFS 7.56 mo	

ID: identification number; TRAEs: treatment-related adverse events; Mo: months; DOR: duration of response; ORR: overall response rate; mPFS: progression-free survival; qd: quaque die (once a day); bid: bis in die (2 times a day).

**Table 2 ijms-25-03304-t002:** Key ongoing clinical trials evaluating KRAS G12C inhibitors, alone or in combination with other compounds.

Study ID/Name	Treatment Regimen	Population
Phase		
Ongoing clinical trials evaluating well established anti-KRAS G12C in combination with other compounds		
NCT04975256/KRYSTAL 14	Adagrasib (MRTX849) + BI 1701963 (inhibitor of KRAS and SOS1 interaction)	advanced *KRAS* G12C mutant solid tumors
Phase I/Ib		
NCT05578092	Adagrasib (MRTX849) + MRTX0902 (SOS1 inhibitor)	advanced solid tumors *KRAS* G12C mutant or harboring any mutations in MAPK pathway effectors
Phase I/II		
NCT05178888/KRYSTAL-16	Adagrasib (MRTX849) + palbociclib	advanced *KRAS* G12C mutant solid tumors
Phase I/Ib		
NCT04330664/KRYSTAL-2	Adagrasib (MRTX849) + TNO155 (SHP2 inhibitor)	advanced *KRAS* G12C mutant solid tumors
Phase I/II		
NCT04793958/KRYSTAL-10	Adagrasib (MRTX849) + cetuximab vs chemotherapy	advanced *KRAS* G12C mutant mCRC
Phase III		
NCT05722327	Adagrasib (MRTX849) + cetuximab and irinotecan	advanced *KRAS* G12C mutant mCRC
Phase I		
NCT06024174	Adagrasib (MRTX849) + BMS-986466 (SHP2 Inhibitor) +/− cetuximab	advanced *KRAS* G12C mutant NSCLC, PDCA, BTC and CRC
Phase 1/2		
NCT04418661	Adagrasib (MRTX849) + RMC-4630 (SHP2 inhibitor)	advanced *KRAS* G12C mutant solid tumors
Phase I		
NCT04892017	DCC-3116 (ULK inhibitor) +/− trametinib, binimetinib, or sotorasib (AMG510)	advanced solid tumors harboring any mutation in RAS/MAPK pathway
Phase I/II		
NCT05480865/Argonaut	Sotorasib (AMG510) + BBP-398 (SHP2 inhibitor)	advanced *KRAS* G12C mutant solid tumors
Phase I		
NCT04929223/INTRINSIC	Divarasib (GDC-6036) + Cetuximab +/− FOLFOX or FOLFIRI (in *KRAS* G12C)	advanced mutant mCRC
Phase I/Ib		
NCT05497336	Fulzerasib (IBI351) + cetuximab (phase Ib) and versus SOC (phase III in mCRC)	advanced *KRAS* G12C mutant solid tumors (phase Ib)
Phase Ib/III		pretreated *KRAS* G12C mutant mCRC (phase III)
NCT06166836	Garsorasib (D-1553) + ifebemtinib (IN10018) (FAK inhibitor)	advanced *KRAS* G12C mutant solid tumors
Phase Ib/II		
Ongoing clinical trials evaluating other anti-KRAS G12C inhibitors		
NCT04165031	LY3499446 +/− several compounds, based on histology (cetuximab in mCRC)	advanced *KRAS* G12C mutant solid tumors
Phase I/II		
NCT 04956640/LOXO-RAS-2000	LY3537982 +/− several compounds, based on histology (cetuximab in mCRC)	advanced *KRAS* G12C mutant solid tumors
Phase I/II		
NCT04699188/KontRASt-01	Opnurasib (JDQ443) +/− TNO155 (SHP2 inhibitor) + tislelizumab	advanced *KRAS* G12C mutant solid tumors
Phase Ib/II		
NCT05358249/KontRASt-03	Opnurasib (JDQ443) + cetuximab (in mCRC)	advanced *KRAS* G12C mutant solid tumors
Phase Ib/II		
NCT05002270	Glecirasib (JAB-21822)	advanced *KRAS* G12C mutant solid tumors
Phase I/II		
NCT05194995	Glecirasib (JAB-21822) + cetuximab	advanced *KRAS* G12C CRC, small intestine cancer and appendiceal cancer
Phase Ib/II		
NCT05288205	Glecirasib (JAB-21822) + JAB-3312 (SHP2 inhibitor)	advanced *KRAS* G12C mutant solid tumors
Phase I/IIa		
NCT04006301	JNJ-74699157	advanced *KRAS* G12C mutant solid tumors
Phase I		
NCT05462717	RMC-6291	advanced *KRAS* G12C mutant solid tumors
Phase I		
NCT06128551	RMC-6291 + RMC-6236 (pan-RAS inhibitor)	advanced *KRAS* G12C mutant solid tumors
Phase Ib		
NCT06117371	BEBT-607	advanced *KRAS* G12C mutant solid tumors
Phase I/Ib		
NCT06006793	SY-5933	advanced *KRAS* G12C mutant solid tumors
Phase I		
NCT06006793	BPI-421286	advanced *KRAS* G12C mutant solid tumors
Phase I		
NCT04973163	BI 1823911	advanced *KRAS* G12C mutant solid tumors
Phase Ia/Ib		
NCT05410145	D3S-001	advanced *KRAS* G12C mutant solid tumors
Phase I		
NCT05768321	GEC255	advanced *KRAS* G12C mutant solid tumors
Phase I		
NCT05485974	HBI-2438	advanced *KRAS* G12C mutant solid tumors
Phase I		

FAK: focal adhesion kinase; SHP: Src homology region 2-containing protein tyrosine phosphatase 2; SOS1: Son of Sevenless homolog 1.

**Table 3 ijms-25-03304-t003:** Summary of the most relevant differences between adagrasib, sotorsaib, and divarsib.

Main Pharmacokinetics Characteristic of KRAS G12C Inhibitors			
Name	Sotorasib (AMG510)	Adagrasib (MRTX849)	Divarasib (GDC-6036)
Mechanism of action	covalent inhibitor of KRAS G12C	covalent inhibitor of KRAS G12C	covalent inhibitor of KRAS G12C
Half-life (hours)	5.5 ± 1.8	24	17.6 ± 2.7
Dose	960 mg qd	600 mg bid	400 mg qd
Median time to maximum concentration (hours)	1	6	2
Other features		CNS penetration	

qd: quaque die (once a day); bid: bis in die (2 times a day): CNS: central nervous system.

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
