# Peer review of "Targeting KRAS G12C Mutation in Colorectal Cancer, A Review: New Arrows in the Quiver"

_ijms, 2024, doi:10.3390/ijms25063304_

Round 1

Reviewer 1 Report

Comments and Suggestions for Authors

Although KRAS G12C mutation is relatively rare in metastatic colorectal cancer (mCRC), however, it results in a worse survival of patients compared to other genetic events. Thus, Ros J et al review an important scientific and clinical problem in this manuscript. The paper is well written, logically structured and easy to follow. However, I have some suggestions.

1. Lines 39-41: a short desciption of the targets of regorafenib, tipiracil etc would help the reader.

2. Lines 53-65: G12D is much more frequent in mCRC than G12C. Could you please briefly compare the mechanism of action of these two mutations? How do they function (differencies)? Why is G12C easier to target than G12D?

3. Figure 1: the arrow at Inavolisib is wrongly edited.

4. Could you please insert a table comparing sotorasib, adagrasib, divarasib? How do they differ in their structure, mechanism, stability etc?

5. Line 338: phosphor-ERK should be corrected to phospho-ERK.

Author Response

Dear Editor and reviewers, 

First of all, we would like to thank for your comments and feedback. All the corrections have done. Please find attached a point-by-point explanation:

1. Lines 39-41: a short desciption of the targets of regorafenib, tipiracil etc would help the reader. Thank you for the commentary, it has been added

2. Lines 53-65: G12D is much more frequent in mCRC than G12C. Could you please briefly compare the mechanism of action of these two mutations? How do they function (differencies)? Why is G12C easier to target than G12D? a brief paragraph regarding how G12D mutation is activated compared to G12C has been added. 

3. Figure 1: the arrow at Inavolisib is wrongly edited. Thank you for pointing out. Figure 1 has been corrected. 

4. Could you please insert a table comparing sotorasib, adagrasib, divarasib? How do they differ in their structure, mechanism, stability etc? Thank you for the suggestion. Table 3 has been added. 

5. Line 338: phosphor-ERK should be corrected to phospho-ERK. Thank you. This type has been corrected

Reviewer 2 Report

Comments and Suggestions for Authors

In this paper, the oncogenic mechanism, clinical drug treatment and drug resistance of KRAS G12C mutation-associated colorectal cancer are reviewed. These would be suitable for publication in IJMS once the below suggestions have been taken into account.

1. Before presenting the clinical data, the authors briefly describe the constitutive activation mechanism of KRAS G12C mutation and the resulting structural pocket for drug targeting. Given that this is a G12C mutation-focused review, a more detailed and illustrated description would be more appropriate.

2. Many studies have pointed to the role of gut microbiota in CRC treatment, it would be interesting to know whether any of them were directly related to the KRAS G12C mutation?

3. It is noted that the authors mentioned the non-genetic resistance mechanism in the treatment of KRAS G12C mutation CRC, could these be related to some plasticity alteration such as altered epigenetic modifications? It may add value to this paper.

Author Response

Dear Editor, dear reviewers,

Thank you for your comments and valuable feedback. I attached a point-by-point explanation of the corrections: 

  1. Before presenting the clinical data, the authors briefly describe the constitutive activation mechanism of KRAS G12C mutation and the resulting structural pocket for drug targeting.Given that this is a G12C mutation-focused review, a more detailed and illustrated description would be more appropriate. Thank you for your comment. Some additional information about the G12C mutation has been added. 
  2. Many studies have pointed to the role of gut microbiota inCRC treatment, it would be interesting to know whether any of them were directly related to the KRAS G12C mutation? Thank you for your comment. We have added a paragraph about microbiome and KRAS mutations. 
  3. It is noted that the authors mentioned the non-genetic resistance mechanism in the treatment of KRAS G12C mutationCRC, could these be related to some plasticity alteration such as altered epigenetic modifications? It may add value to this paper. Thank you so much for noticing this. A recent paper in lung cancer demonstrated that histological switch is produced by epigenetic modifications. This has been added
